# Sampling-Decomposable Generative Adversarial Recommender

**Binbin Jin**[†], **Defu Lian**[†§*], **Zheng Liu**[‡], **Qi Liu**[†§] **Jianhui Ma**[†], **Xing Xie**[‡] , **Enhong Chen**[†§*]
[†]School of Computer Science and Technology, University of Science and Technology of China
[§]School of Data Science, University of Science and Technology of China
[‡]Microsoft Research Asia
bb0725@mail.ustc.edu.cn, {liandefu, qiliuql, jianhui, cheneh}@ustc.edu.cn,
{zhengliu, xingx}@microsoft.com

## Abstract

Recommendation techniques are important approaches for alleviating information overload. Being often trained on implicit user feedback, many recommenders suffer from the sparsity challenge due to the lack of explicitly negative samples. The GAN-style recommenders (i.e., IRGAN) addresses the challenge by learning a generator and a discriminator adversarially, such that the generator produces increasingly difficult samples for the discriminator to accelerate optimizing the discrimination objective. However, producing samples from the generator is very time-consuming, and our empirical study shows that the discriminator performs poor in top-k item recommendation. To this end, a theoretical analysis is made for the GAN-style algorithms, showing that the generator of limit capacity is diverged from the optimal generator. This may interpret the limitation of discriminator's performance. Based on these findings, we propose a Sampling-Decomposable Generative Adversarial Recommender (SD-GAR). In the framework, the divergence between some generator and the optimum is compensated by self-normalized importance sampling; the efficiency of sample generation is improved with a sampling-decomposable generator, such that each sample can be generated in $O(1)$ with the Vose-Alias method. Interestingly, due to decomposability of sampling, the generator can be optimized with the closed-form solutions in an alternating manner, being different from policy gradient in the GAN-style algorithms. We extensively evaluate the proposed algorithm with five real-world recommendation datasets. The results show that SD-GAR outperforms IRGAN by 12.4% and the SOTA recommender by 10% on average. Moreover, discriminator training can be 20x faster on the dataset with more than 120K items.

## 1 Introduction

With the popularity of Web 2.0, content disseminated on Internet has been growing explosively, which greatly intensifies the information overload problem. Recommender system is regarded as an important approach to address this problem by filtering out irrelevant information automatically. In recent years, personalized ranking algorithms, such as [7, 11, 16, 17, 13, 18, 32], have been widely used in E-commerce and online advertisement, creating huge business value and social impact for various kinds of web services. Since the recommendation algorithms are usually trained with implicit user feedback such as click and purchase history, the lack of explicitly negative samples become an imperative problem. In other words, how to discover and utilize informative negative samples becomes critical in optimizing the learning performance.

---

[*]Defu Lian and Enhong Chen are corresponding authors.

Existing work on negative sampling can be grouped into two categories. One category is to treat all items without user interaction as negative samples, which are assigned with a small confidence score [12]. The algorithm has been proved to impose the gravity regularizer, which penalizes the non-zero prediction for uninteracted items [1]. A number of algorithms have been developed for its optimization, from batch-based ALS to mini-batch SGD [15]. To further distinguish the confidence of being negative, the user-item confidence matrix has been regularized to be sparse and low-rank to facilitate learning efficiency [24, 8]. The other category is to sample negative items from those uninteracted ones with various kinds of neural networks such as GAN-based models [4, 33], GCN-based models [9] and AE-based models [17]. A widely used sampling strategy is to draw negative items either w.r.t. uniform distribution [25, 37, 29], popularity [23], or based on recommendation models [26, 36, 11, 34]. Sampling based on recommendation models is regarded to be more effective; and in recent years, the GAN-style algorithms, e.g., IRGAN [34], become highly popular. It is also discussed in [6], that the GAN-style framework could be a promising direction of discovering informative negative samples. As discussed in this work, it is also more possible for the framework to seamlessly integrate approximate search algorithms, such as ALSH [28, 21], PQ [14] and HNSW [20], with complex recommendation algorithms, so as to mutually reinforce the learning of both algorithms.

The GAN-style recommendation algorithms learn a generator and a discriminator in an iterative and adversarial way, such that the generator may produce increasingly difficult samples for the discriminator to accelerate optimizing the discrimination objective. However, the existing GAN-style recommendation algorithms may suffer from two severe limitations. On the one hand, the discriminator rather than the generator is more suitable for the top-k item recommendation [5, 3]. In addition to the fact that the generator acts as a negative sampler, another reason is that the discriminator learns directly from training data, whereas the generator merely learns from samples drawn from the generator distribution; besides, the learning of the generator is guided by the discriminator, which can be not reliable. Unfortunately, the discriminator of existing algorithms performs very poor according to our empirical study. On the other hand, generating samples from the generator is time-consuming due to the generator's large sample space, which restricts it from being applied for large-scale datasets.

To this end, in this paper, a theoretical analysis is made for the GAN-style algorithms, which shows that a generator of enough capacity has the optimal solution given the discriminator. The divergence between the generator of limit capacity and the optimum may lead to the limitation of discriminator's performance. Based on these findings, we propose the Sampling-Decomposable Generative Adversarial Recommender (SD-GAR), where sampling is carried out with a decomposable generator. In the framework, the divergence between the generator and its optimum is compensated by self-normalized importance sampling, so that the recommendation performance of discriminator is considerably improved. Another interesting result of using the self-normalized importance sampling is that if the generator is degenerated to the uniform distribution, the discriminator intrinsically subsumes recommendation models with dynamic negative sampling [37] and self-adversarial sampling [29]. Due to decomposability of sampling, the generator can be optimized with its closed-form solutions in an alternating manner, which is different from using policy gradient as in the GAN-style algorithms and can lead to better training efficacy. More importantly, the efficiency of sample generation can be remarkably improved, where each sample can be generated in $O(1)$ with the Vose-Alias method [31]. We extensively evaluate the proposed algorithm with five real-word recommendation datasets of varied size and difficulty of recommendation. The experimental results show that the proposed algorithm outperforms IRGAN by 12.4% on average and the SOTA recommender by 10% w.r.t. NDCG@50. The efficiency study indicates that discriminator training can be accelerated by 20x in the dataset with more than 120K items.

## 2 Sampling-Decomposable Generative Adversarial Recommender

Before elaborating the proposed SD-GAR, we first briefly introduce the GAN-style recommenders (e.g., IRGAN [34]) and provide some theoretical analysis results. Following that, we propose a new objective function base on expectation approximation with self-normalized importance sampling. Within the objective function, we propose a sampling-decomposable generator and investigate its optimization algorithm. Finally, we provide complexity analysis to SD-GAR and comparison with IRGAN. In the following, to be generic, we use context to represent user, time, location, behavior history and so on. Denote by $\mathcal{C}$ the set of $N$ contexts, $\mathcal{I}$ the set of $M$ items and $\mathcal{I}_c$ interacted items in a context $c$.

## 2.1 Analysis to IRGAN

Generally speaking, IRGAN applies a game theoretical minimax game in the GAN framework for information retrieval (IR) tasks, and has been used for three specific IR scenarios including web search, item recommendation and question answering. More specifically, IRGAN iteratively optimizes a generator $G$ and a discriminator $D$, such that the generator produces increasingly difficult samples for the discriminator to minimize the discrimination objective. In case of recommendation from implicit user feedback, IRGAN formally optimizes the following objective function:

$$\max_G \min_D \mathcal{J}(D, G) = \sum_{c \in \mathcal{C}} -\mathbb{E}_{i \sim P_{\text{true}}(\cdot|c)} \log D(i|c) - \mathbb{E}_{j \sim P_G(\cdot|c)} \log \left(1 - D(j|c)\right), \tag{1}$$

where $P_{\text{true}}(\cdot|c)$ is an underlying true relevance distribution over candidate items and $P_G(\cdot|c)$ is a probability distribution used to generate negative samples. $D(i|c) = \sigma(g_\phi(c, i)) = \frac{1}{1+\exp(-g_\phi(c,i))}$ estimates the probability of preferring item $i$ in a context $c$. As generative process is stochastic process over discrete data, IRGAN applies the REINFORCE algorithm for optimizing the generator. When $D$ and $G$ is well trained, the generator $G$ is used for recommendation, since the discriminator $D$ performs poor in practice according to our empirical study. However, by considering the generator as a negative sampler so as to address the sparsity issue, the discriminator $D$ rather than $G$ should be used for recommendation. Before understanding this problem, we first provide theoretical analysis for IRGAN.

**Theorem 2.1.** *Assume $G$ has enough capacity. Given the discriminator $D$, $\min_G \mathcal{J}(D, G)$ achieves the optimum when*

$$P_{G^\star}(\cdot|c) = \text{one-hot}(\arg\max_i(g_\phi(c, i))). \tag{2}$$

The proof is provided in the appendix. When generating samples from $P_{G^*}$, it is reduced to binary function search problem [30], and many approximate search algorithms, such as PQ, HNSW and ALSH, can be applied for fast search. Therefore, this framework seamlessly integrates approximate search algorithms with any complex recommendation algorithms, which are represented by the discriminator $D$. However, sampling from $P_{G^\star}$ suffers from the false negative issue, since items with large $g_\phi(c, i)$ score are also more likely to be positive. To this end, we introduce randomness into the generator by imposing entropy regularization over the generator, i.e., $\mathcal{H}(P_G(\cdot|c)) = -\sum_{i \in \mathcal{I}} P_G(i|c) \log P_G(i|c)$.

**Theorem 2.2.** *Assume $G$ has enough capacity and let $f_c(i) = -\log(1 - D(i|c)) = \log(1 + \exp(g_\phi(c, i)))$. Given the discriminator $D$, $\min_G \mathcal{J}(D, G) - T \cdot \mathcal{H}(P_G(\cdot|c))$ achieves the optimum when*

$$P_{G_T^\star}(i|c) = \frac{\exp\left(f_c(i)/T\right)}{\sum_{j \in \mathcal{I}} \exp\left(f_c(j)/T\right)}. \tag{3}$$

The proof is provided in the appendix. When $T \to 0$, $P_{G_T^\star} \to P_{G^\star}$, and when $T \to +\infty$, $P_{G_T^\star}$ is degenerated to a uniform distribution. Therefore, $T$ controls the randomness of the generator.

## 2.2 A New Objective Function

It is infeasible to directly sample from $P_{G_T^\star}$ due to large sample space and potentially large computational cost of $g_\phi(c, i)$. Therefore, a generator $Q_G(\cdot|c)$ with limit capacity is usually used to approximate the $P_{G_T^\star}$, with the aim of improving sampling efficiency. However, the divergence between the generator with limit capacity and the optimum with sufficient capacity may lead to the limitation of discriminator's performance in top-k item recommendation. To compensate the divergence between them, we propose to approximate $\mathcal{J}(D, G_T^\star)$ with self-normalized importance sampling, by drawing a set of samples $\mathcal{S}_c$ from $Q_G(\cdot|c)$ in each context. Formally, for each context, given a set of samples $\mathcal{I}_c$ from $P_{\text{true}}(\cdot|c)$ observed as the training data, $\mathcal{J}(D, G_T^\star)$ is approximated as

$$\mathcal{J}(D, G_T^\star) \approx \mathcal{V}_T(D, \mathcal{S}) = \sum_{c \in \mathcal{C}} \left( -\frac{1}{|\mathcal{I}_c|} \sum_{i \in \mathcal{I}_c} \log D(i|c) - \sum_{j \in \mathcal{S}_c} w_{cj} \log\left(1 - D(j|c)\right) \right),$$

$$w_{cj} = \frac{\exp\left(f_c(j)/T - \log \tilde{Q}_G(j|c)\right)}{\sum_{i \in \mathcal{S}_c} \exp\left(f_c(i)/T - \log \tilde{Q}_G(i|c)\right)}. \tag{4}$$

where $\mathcal{S} = \bigcup_{c \in \mathcal{C}} \mathcal{S}_c$ and $\tilde{Q}_G(j|c)$ is the unnormalized $Q_G(j|c)$. The detailed derivation is provided in the appendix. This approximate quantity satisfies the following properties.

**Proposition 2.1** (**Theorem 9.2** [22]). $\mathcal{V}_T(D, \mathcal{S})$ *is an asymptotic unbiased estimator of* $\mathcal{J}(D, G_T^\star)$,

$$\mathbb{P}\left(\lim_{\forall c, |\mathcal{S}_c| \to \infty} \mathcal{V}_T(D, \mathcal{S}) = \mathcal{J}(D, G_T^\star)\right) = 1.$$

Since we focus on modeling $Q_G(\cdot|c)$ rather than $P_{\text{true}}(\cdot|c)$ in this paper, we only consider the uncertainty from $Q_G(\cdot|c)$ which can provide guidance for variance reduction and help optimize the proposal $Q_G(\cdot|c)$. The variance is then approximated with the delta method [22],

$$\text{Var}\left(\mathcal{V}_T(D, \mathcal{S})\right) = \sum_{c \in \mathcal{C}} \frac{1}{|\mathcal{S}_c|} \sum_{i \in \mathcal{I}} \frac{P_{G_T^\star}(i|c)^2 (f_c(i) - \mu_c)^2}{Q_G(i|c)}, \tag{5}$$

where $\mu_c = \mathbb{E}_{i \sim P_{G_T^\star}(\cdot|c)}(f_c(i))$. It is not possible to approach 0 variance with ever better choices of $Q_G$ due to the following proposition.

**Proposition 2.2.** *Var* $\left(\mathcal{V}_T(D, \mathcal{S})\right) \geq \sum_{c \in \mathcal{C}} \frac{1}{|\mathcal{S}_c|} \mathbb{E}_{i \sim P_{G_T^\star}(\cdot|c)}(|f_c(i) - \mu_c|)^2$, *where the equality holds if* $Q_G(i|c) \propto P_{G_T^\star}(i|c)|f_c(i) - \mu_c|$.

The proof is provided in the appendix. The result can provide guidance for designing better learning algorithms of $Q_G$.

In addition to these properties, this new objective function is also quite a generic framework to integrate any negative samplers with recommendation algorithms, which can subsume several existing algorithms when the generator is degenerated to the uniform distribution.

**Proposition 2.3.** *If* $\forall c$, $\mathcal{S}_c$ *drawn i.i.d from uniform*$(\mathcal{I})$, *then* $w_{cj} = \frac{\exp(f_c(j)/T)}{\sum_{i \in \mathcal{S}_c} \exp(f_c(i)/T)}$ *and* $\forall 0 < T_1 < T_2 < +\infty$,

$$\lim_{T \to +\infty} \mathcal{V}_T(D, \mathcal{S}) < \mathcal{V}_{T_2}(D, \mathcal{S}) < \mathcal{V}_{T_1}(D, \mathcal{S}) < \lim_{T \to 0} \mathcal{V}_T(D, \mathcal{S}).$$

The proof is provided in the appendix. Note that $\lim_{T \to 0} \mathcal{V}_T(D, \mathcal{S})$ corresponds to logit-loss based recommender with dynamic negative sampling [37], and $\lim_{T \to +\infty} \mathcal{V}_T(D, \mathcal{S})$ corresponds to logit-loss based recommender with uniform sampling [25, 19]. The recently proposed self-adversarial loss [29] is also a special case, by simply preventing gradient propagating through sample importance.

## 2.3 Sampling-Decomposable Generator

In most GAN-style recommendation algorithms, though the generator is of limit capacity, sampling from the generator is still time-consuming due to the requirement of on-the-fly probability computation. To this end, we propose a sampling-decomposable generator, where sampling from the generator is decomposed into two steps. The first step is to sample from a probability distribution over latent states conditioned on contexts, and the second step is to sample from a probability distribution over $M$ candidate items conditioned on latent states. Formally, assuming $\boldsymbol{x}_c$ defines the probability distribution over $K$ latent states and $\boldsymbol{y}_{\cdot k}$ defines the probability distribution over candidate items, then the generator is decomposed as

$$Q_G(\cdot|c) = \sum_{k=0}^{K-1} x_{c,k} \boldsymbol{y}_{\cdot k} = \boldsymbol{Y} \boldsymbol{x}_c, \tag{6}$$

where $\boldsymbol{Y} = [\boldsymbol{y}_{\cdot 0}, \cdots, \boldsymbol{y}_{\cdot K-1}]$ stacks $\boldsymbol{y}_{\cdot k}$ by column, being subject to $\mathbf{1}_M^\top \boldsymbol{Y} = \mathbf{1}_K$. Therefore, it is easy to verify that such a sampling-decomposable generator satisfies $\sum_{i \in \mathcal{I}} Q_G(i|c) = 1$. The probability decomposablity has been widely used in probabilistic graphical models, such as PLSA [10] and LDA [2]. To draw samples from the generator, we can first sample a latent state $k$ from the distribution $\boldsymbol{x}_c$, and then draw an item from the distribution $\boldsymbol{y}_{\cdot k}$. Since the probability tables only occupy $O((M + N)K)$ space, we also pre-compute the alias table for $\boldsymbol{x}_c$ and $\boldsymbol{y}_{\cdot k}$ according to the Vose-Alias method. As a consequence, due to the sampling-decomposable assumption, each item can be generated from the generator in $O(1)$, achieving remarkable speedup for item sampling.

## 2.4 Optimization of Generator

Though the sampling-decomposable assumption decreases the model capacity, the generator is not necessarily optimized by the REINFORCE algorithm any more, so that better training efficacy may be achieved. Following IRGAN and trying to reduce the variance of the objective estimator $\mathcal{V}_T(D, \mathcal{S})$ according to Proposition 2.2, we propose the follow optimization problem to learning the generator,

$$\max_{\boldsymbol{X} \geq 0, \boldsymbol{Y} \geq 0} \sum_{c \in \mathcal{C}} \sum_{i \in \mathcal{I}} \boldsymbol{x}_c^\top \boldsymbol{y}_i P_{G_T^\star}(i|c) |f_c(i) - \mu_c|, \text{ s.t. } \boldsymbol{X} \mathbf{1}_K = \mathbf{1}_N \text{ and } \mathbf{1}_M^\top \boldsymbol{Y} = \mathbf{1}_K, \tag{7}$$

where $\boldsymbol{X} = [\boldsymbol{x}_0, \cdots, \boldsymbol{x}_{N-1}]^\top$ stacks $\boldsymbol{x}_c$ by row. To solve this problem, alternating optimization can be applied, which iteratively update $\boldsymbol{X}$ and $\boldsymbol{Y}$ until convergence. In particular, when $\boldsymbol{Y}$ fixed, the optimization problem w.r.t. $\boldsymbol{x}_c$ is then formulated as

$$\max_{\boldsymbol{x}_c \in \mathcal{A}_K} \boldsymbol{x}_c^\top \sum_{i \in \mathcal{I}} \boldsymbol{y}_i P_{G_T^\star}(i|c) |f_c(i) - \mu_c| + \lambda_X \mathcal{H}(\boldsymbol{x}_c), \tag{8}$$

where $\mathcal{A}_K = \{\boldsymbol{a} \in \mathbb{R}_+^K | \boldsymbol{a}^\top \mathbf{1}_K = 1\}$ the $(K-1)$-simplex and $\mathcal{H}(\boldsymbol{x}_c) = -\sum_k x_{c,k} \log x_{c,k}$ is an entropy regularizer, to prevent $\boldsymbol{x}_c$ from degenerating to the one-hot distribution. Based on Theorem 2.2, the solution for the problem (8) is derived as follows.

**Corollary 2.1.** *Let* $\boldsymbol{b}_c = \sum_{i \in \mathcal{I}} \boldsymbol{y}_i P_{G_T^\star}(i|c) |f_c(i) - \mu_c|$. *The objective function in Eq* (8) *achieves the optimum when*

$$\boldsymbol{x}_c = softmax(\boldsymbol{b}_c / \lambda_X). \tag{9}$$

To obtain $\boldsymbol{x}_c$, it is required to first calculate these quantities $\mu_c$ and $\boldsymbol{b}_c$. It is infeasible to straightforwardly compute them since they involve summation over $M$ items. Therefore, we again resort to self-normalized importance sampling to perform approximate estimation of these two expectations. To reduce bias of the estimators, we draw different sample sets from the most recent proposal $Q_G(i|c) = \boldsymbol{x}_c^\top \boldsymbol{y}_i$ for separate use.

When $\boldsymbol{X}$ fixed, the optimization problem w.r.t. $\boldsymbol{y}_{\cdot k}$, the $k$-th column of $\boldsymbol{Y}$, is then formulated as:

$$\max_{\boldsymbol{y}_{\cdot k} \in \mathcal{A}_M} \sum_{i \in \mathcal{I}} y_{i,k} \sum_{c \in \mathcal{C}} x_{c,k} P_{G_T^\star}(i|c) |f_c(i) - \mu_c| + \lambda_Y \mathcal{H}(\boldsymbol{y}_{\cdot k}). \tag{10}$$

where $\mathcal{H}(\boldsymbol{y}_{\cdot k})$ is an entropy regularizer, to prevent $\boldsymbol{y}_{\cdot k}$ from degenerating to the one-hot distribution. Following the Theorem 2.2, we can derive the solution of the problem as follows.

**Corollary 2.2.** *Let* $d_{k,i} = \sum_{c \in \mathcal{C}} x_{c,k} P_{G_T^\star}(i|c) |f_c(i) - \mu_c|$. *The objective function in Eq* (10) *achieves the optimum when*

$$\boldsymbol{y}_{\cdot k} = softmax(\boldsymbol{d}_k / \lambda_Y). \tag{11}$$

To obtain $\boldsymbol{y}_{\cdot k}$, it is required to calculate $\mu_c$ and $\boldsymbol{d}_k$. After $\boldsymbol{x}_c$ is updated, $\mu_c$ is re-estimated by drawing a new sample set from the updated proposal distribution. To estimate $\boldsymbol{d}_k$, we first use self-normalized importance sampling to estimate the normalize constant $Z_{G_T^\star}(c)$ of $P_{G_T^\star}(i|c)$. Then we use the probability $Q_G(c|i) = \sum_k P(k|i) P(c|k)$ for context sampling, where $P(k|i) = \frac{y_{i,k}}{\sum_k y_{i,k}}$ and $P(c|k) = \frac{x_{c,k}}{\sum_c x_{c,k}}$ by assuming the prior $P(k) = \frac{1}{K}$ and $P(c) = \frac{1}{N}$. With the Vose-Alias method, we draw a sample set of contexts $\mathcal{S}_i$ for approximating $d_{k,i}$ as follows:

$$d_{k,i} = \mathbb{E}_{c \sim Q_G(c|i)} \frac{P_{G_T^\star}(i|c)}{Q_G(c|i)} x_{c,k} |f_c(i) - \mu_c| \approx \frac{1}{|\mathcal{S}_i|} \sum_{c \in \mathcal{S}_i} \hat{w}_{ic} x_{c,k} |f_c(i) - \mu_c|. \tag{12}$$

where $\hat{w}_{ic} = \exp\left(f_c(i)/T - \log Q_G(c|i) - \log \tilde{Z}_{G_T^\star}(c)\right)$.

Algorithm 1 shows the overall procedure of iteratively updating the discriminator and the generator, where the parameters of the generator are randomly initialized.

## 2.5 Time Complexity Analysis

Thanks to our sampling-decomposable generator and Vose-Alias method sampling techniques, generating an item from the generator only requires $O(1)$. Therefore, if $|\mathcal{S}_c|$ is a multiplier of $|\mathcal{I}_c|$, the

---

**Algorithm 1:** Sampling-Decomposable Generative Adversarial Recommender

---

**Input:** Context set $\mathcal{C}$; interacted items set $\mathcal{I}_c$; sample sets $\mathcal{S}_c, \mathcal{S}_i$; temperature $T, \lambda_X, \lambda_Y$; total iterations $L$; update frequency $l_g$ of the generator
**Output:** Parameters $\Theta$ of the discriminator

1  $\boldsymbol{X} \sim U(0,1)$;
2  $\boldsymbol{Y} \sim U(0,1)$;
3  **foreach** $c \in \mathcal{C}$ **do**
4      $P(k|c) \leftarrow \text{AliasTable}(\boldsymbol{x}_c / \boldsymbol{x}_c^\top \mathbf{1}_K)$;          // $O(K)$
5  **for** $k = 0...K-1$ **do**
6      $P(i|k) \leftarrow \text{AliasTable}(\boldsymbol{y}_{\cdot k} / \boldsymbol{y}_{\cdot k}^\top \mathbf{1}_M)$;          // $O(M)$
7  **for** *l=1...L* **do**
8      **foreach** *interacted item set* $\mathcal{I}_c$ **do**
9         draw item sample set $\mathcal{S}_c$ as negatives based on $P(k|c)$ and $P(i|k)$;      // $O(|\mathcal{S}_c|)$
10        update parameters $\Theta$ of the discriminator by minimizing Eq (4);
11     **if** *l % $l_g$ == 0* **then**
12        **foreach** $c \in \mathcal{C}$ **do**
13           draw item sample set $\mathcal{S}_c$ based on $P(k|c)$ and $P(i|k)$;      // $O(|\mathcal{S}_c|)$
14           compute $\boldsymbol{x}_c$ using Eq (9);
15           $P(k|c) \leftarrow \text{AliasTable}(\boldsymbol{x}_c)$;          // $O(K)$
16        **for** $k = 0...K-1$ **do**
17           $P(c|k) \leftarrow \text{AliasTable}(\boldsymbol{x}_{\cdot k} / \boldsymbol{x}_{\cdot k}^\top \mathbf{1}_N)$;      // $O(N)$
18           $P(k|i) \leftarrow \text{AliasTable}(\boldsymbol{y}_i / \boldsymbol{y}_i^\top \mathbf{1}_K)$;      // $O(K)$
19           draw context sample set $\mathcal{S}_i$ for each item based on $P(c|k)$;      // $O(|\mathcal{S}_i|)$
20           compute $\boldsymbol{y}_{\cdot k}$ using Eq (11);
21           $P(i|k) \leftarrow \text{AliasTable}(\boldsymbol{y}_{\cdot k})$;      // $O(M)$
22 **return** $\Theta$

---

time complexity of training the discriminator is still linearly proportional to the data size. When learning parameters $\boldsymbol{X}$ and $\boldsymbol{Y}$ of the generator, the time complexity of the proposed algorithm is $O(NK|\mathcal{S}_c| + MK|\mathcal{S}_i|)$, where $|\mathcal{S}_c|$ is the size of the item sample set for approximation and $|\mathcal{S}_i|$ is the size of the context sample set for approximation. The value of $|\mathcal{S}_c|$ and $|\mathcal{S}_i|$ is usually very small compared to the number of items $M$ and the number of contexts $N$. Therefore, training the generator is very efficient. Moreover, we empirically find that lowering the frequency of updating the generator would almost not affect the overall recommendation performance, so we choose to update the generator every $l_g$ iterations.

**Comparison with IRGAN**. SD-GAR is similar to IRGAN. However, according to [34], the time complexity of training IRGAN is $O(NMK)$, where $O(K)$ indicates the time cost of computing preference score for each item. Therefore, the proposed algorithm is much more efficient than IRGAN.

## 3 Experiments

We first compare our proposed SD-GAR with a set of baselines, including the state-of-the-art model. Then, we show the efficiency and scalability of SD-GAR from two perspectives.

### 3.1 Datasets

As shown in Table 1, five publicly available real-world datasets [2] are used for evaluating the proposed algorithm. The datasets vary in difficulty of item recommendation, which may be indicated by the

Table 1: Dataset Statistics. Concentration indicates rating percentage on top 5% most popular items.

|  | #users | #items | #ratings | density | concentration |
|---|---|---|---|---|---|
| CiteULike | 7,947 | 25,975 | 134,860 | 6.53e-04 | 31.56% |
| Gowalla | 29,858 | 40,988 | 1,027,464 | 8.40e-04 | 29.15% |
| Amazon | 130,380 | 128,939 | 2,415,650 | 1.44e-04 | 32.98% |
| MovieLens | 60,655 | 8,939 | 2,105,971 | 3.88e-03 | 61.98% |
| Echonest | 217,966 | 60,654 | 4,422,471 | 3.35e-04 | 45.63% |

numbers of items and ratings, the density and concentration. The Amazon dataset is a subset of customers' ratings for Amazon books and the MovieLens dataset is from the classic MovieLens10M dataset. For Amazon and MovieLens10M, we treat items with scores higher than 4 as positive. The CiteULike dataset collects users'personalized libraries of articles, the Gowalla dataset includes users' check-ins at locations, and the Echonest dataset records users' play count of songs. To ensure each user can be tested, we filter these datasets such that users rated at least 5 items. For each user, we randomly sample her 80% ratings into a training set and the rest 20% into a testing test. 10% ratings of the training set are used for validation. We build recommendation models on the training set and evaluate them on the test set.

## 3.2 Experimental Setup

In this paper, our proposed SD-GAR is implemented based on Tensorflow and trained with the Adam algorithm on a linux system (2.10GHz Intel Xeon Gold 6230 CPUs and a Tesla V100 GPU). In addition, there are some hyper-parameters. Note that the discriminator can be any existing model. In this paper, we choose to use the matrix factorization model (i.e., $D(\cdot|c) = \sigma(\boldsymbol{Y} \boldsymbol{x}_c + \boldsymbol{b})$ where $\boldsymbol{b}$ is the bias of items) since it is simple and even superior to neural recommendation algorithms in some cases [27]. Unless otherwise specified, the dimension of user and item embeddings is set to 32. The batch size is fixed to 512 and the learning rate is fixed to 0.001. We impose L2 regularization to prevent overfitting and its coefficient is tuned over $\{0.01, 0.03, 0.05\}$ on the validation set. The number of item sample set for learning the discriminator is set to 5. The number of item and context sample set for learning the generator is set to 64. The temperature $T, \lambda_X, \lambda_Y$ is tuned over $\{0.1, 0.5, 1\}$. The sensitivity of some important parameters is discussed in the appendix.

The performance of recommendation is assessed by how well positive items on the test set are ranked. We exploit the widely-used metric NDCG for ranking evaluation. NDCG at a cutoff k, denoted as NDCG@k, rewards method that ranks positive items in the first few positions of the top-k ranking list. The positive items ranked at low positions of ranking list contribute less than positive items at the top positions. The cutoff k in NDCG is set to 50 by default.

## 3.3 Baselines

To validate the effectiveness of SD-GAR, we compare it against seven popular methods.

- **BPR** [25] is a pioneer work of personalized ranking in recommender systems. It uses pairwise ranking loss and randomly samples negative items.

- **AOBPR** [26] improves BPR with adaptive oversampling. We use the implementation in LibRec [3].

- **WARP** [35] uses the weighted approximate-rank pairwise loss function for collaborative filtering whose loss is based on a ranking error function. We use the implementation in LightFM [4].

- **CML** [11] learns a joint metric space to encode users' preference as well as user-user and item-item similarity and uses WARP loss function for optimization. We use the authors' released code [5]. Following the original paper, the margin size is tuned over $\{0.5, 1.0, 1.5, 2.0\}$.

- **DNS** [37] draws a set of negative samples from a uniform distribution but only leaves one item with the highest predicted score to update the model.

Table 2: Comparison with baselines on five datasets with respect to NDCG@50

|  | CiteULike | | Gowalla | | MovieLens | | Amazon | | Echonest | | |
|---|---|---|---|---|---|---|---|---|---|---|---|
|  | NDCG@50 | imp% | NDCG@50 | imp% | NDCG@50 | imp% | NDCG@50 | imp% | NDCG@50 | imp% | AvgImp |
| BPR | 0.1165 | 17.2 | 0.1255 | 29.1 | 0.2605 | 20.3 | 0.0387 | 78.3 | 0.0882 | 43.7 | 37.7 |
| AOBPR | 0.0977 | 39.8 | 0.1349 | 20.1 | 0.2545 | 23.1 | 0.0573 | 20.6 | 0.1038 | 22.0 | 25.1 |
| WARP | 0.0948 | 44.0 | 0.1397 | 16.0 | 0.2546 | 23.1 | 0.0553 | 24.9 | 0.1116 | 13.5 | 24.3 |
| CML | 0.1194 | 14.4 | 0.1303 | 24.3 | 0.2737 | 14.5 | 0.0572 | 20.9 | 0.1035 | 22.4 | 19.3 |
| DNS | 0.1157 | 18.0 | 0.1412 | 14.7 | 0.2693 | 16.4 | 0.0580 | 19.0 | 0.1013 | 25.0 | 18.6 |
| IRGAN | 0.1174 | 16.2 | 0.1443 | 12.3 | 0.2858 | 9.7 | 0.0627 | 10.2 | 0.1114 | 13.7 | 12.4 |
| SA | 0.1269 | 7.6 | 0.1490 | 8.7 | 0.2764 | 13.4 | 0.0619 | 11.7 | 0.1138 | 11.3 | 10.5 |
| SD-GAR | **0.1365** | - | **0.1620** | - | **0.3134** | - | **0.0691** | - | **0.1267** | - | - |

- **IRGAN** [34] is a state-of-the-art GAN based model including a generative network that generates items for a user and a discriminative network that determines whether the instance is from real data or generated. We use the authors' released code [6].

- **SA** [29] is a special case of the proposed method where the generator is replaced by a uniform distribution and the gradient from sample importance is forbidden.

For the sake of fairness, hyperparameters of these competitors (e.g., the embedding size and the number of negative samples) are set to the same as SD-GAR.

## 3.4 Experimental Results

**Overall Performance.** In this experiment, we show the comparison results between SD-GAR and the baselines in Table 2. In addition to the comparative results, we also analyze some potential limits and effective mechanism of the baselines.

- SD-GAR consistently outperforms all baselines on all five datasets. On average, the relative performance improvements are at least 10.5%. The relative improvements to the classic method BPR reach 37.7% on average. This fully validates the effectiveness of SD-GAR.

- Among the baselines, AOBPR is a method that approximates the rank of items to sampling process while DNS, IRGAN, SA utilize the predicted scores for this goal. According to the experimental results, we can find that AOBPR is not competitive in most cases. This may be because applying predicted scores can distinguish the importance of items at a finer granularity and lead to better results. WARP and CML are two methods utilizing rejection sampling from a uniform distribution. After they are better trained, it is hard to draw informative samples which limit their performance.

- We find that weighting negative samples with predicted scores can bring much improvements. This is based on the observation that SA and DNS outperform BPR. In particular, SA assigns higher weights to items with larger predicted scores, while DNS only leaves the negative item with the highest score. In addition, the fact that SA is superior than DNS also implies introducing more negative samples can improve coverage and performance.

- IRGAN is beaten by SA due to the divergence between the generator and its optimum. By compensating the divergence with self-normalized importance sampling, our proposed SD-GAR obtains 12.4% improvements on average and achieves best performance among all baselines.

- Figure 1(a) shows the performance trends with respect to the number of iterations comparing with three classic methods on the Gowalla dataset. We can find SD-GAR stays ahead of other baselines along with the training process and starts to converge when it comes to around 30 iterations.

**Comparison of Time Consumption.** Here, we illustrate the efficiency of SD-GAR. As shown in Figure 1(b), we compare SD-GAR with IRGAN since they have similar frameworks which contain a generative network and a discriminative network. We give the results on two large-scale datasets (i.e., Amazon, Echonest). In the Figure, the blue bars represent the training time of the discriminator while the red bars represent the training time of the generator. Regarding the discriminator, the training time of SD-GAR is 20x (10x) faster on the Amazon (Echonest) dataset. Regarding the generator,

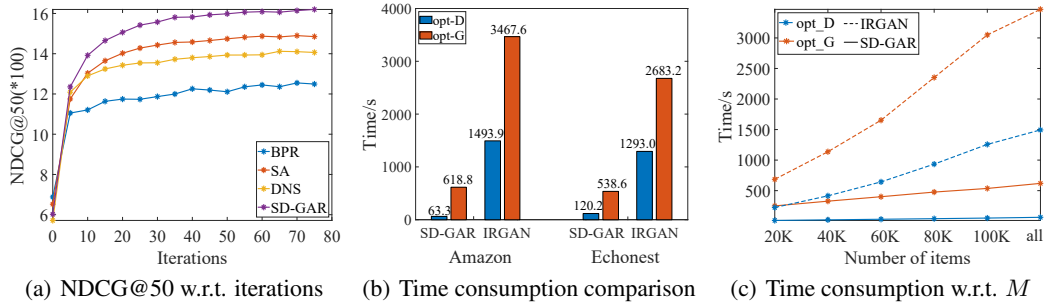

| (a) NDCG@50 w.r.t. iterations | (b) Time consumption comparison | (c) Time consumption w.r.t. $M$ |
|---|---|---|

Figure 1: (a) shows the performance trends between SD-GAR and classic beselines. (b) and (c) show the comparison of time consumption between SD-GAR and IRGAN.

the training time of SD-GAR is 5x faster on both datasets. In addition, note that our generator is optimized at a frequency of $l_g > 1$ iterations so that its training time per iteration is much shorter.

Figure 1(c) shows the time consumption of two models with respect to the number of items. Specifically, we conduct this experiment on the Amazon dataset as it consists of more than 120K items. We random select 20K, 40K, 60K, 80K, 100K items and remove the irrelevant data. In the Figure, the dotted lines are the training time of IRGAN while the solid lines are the training time of SD-GAR. From the results, less time is spent in training the discriminator and generator of SD-GAR. In addition, the time consumption of both models grows linearly with the increasing number of items. However, the growth rate of IRGAN is much larger than that of SD-GAR. Therefore, SD-GAR is much more efficient and scalable than IRGAN.

## 4 Conclusion

In this paper, we proposed a sampling-decomposable generative adversarial recommender, to address low efficiency of negative sampling with the generator and bridge the gap between the generator of limit capacity and the optimal generator with sufficient capacity. The proposed algorithm was evaluated with five real-world recommendation datasets, showing that the proposed algorithm significantly outperforms the competing baselines, including the SOTA recommender. The efficiency study showed that the training of the proposed algorithm achieves remarkable speedup. The future work includes the better design and the learning mechanism of sampling-efficient generators.

## Broader Impact

In this paper, we develop a new recommendation algorithm, which aims to efficiently solve the sparsity challenge in recommender system. The offline evaluation results on multiple datasets show that the new algorithm achieves better recommendation performance in terms of NDCG. The task does not leverage any biases in the data. As a consequence, the customers who often use recommendation services may more easily figure out their interested products, the researchers who design new recommendation algorithms may be inspired by the insight delivered in this paper, and the engineers who develop recommendation algorithms may implement the new algorithm and incorporate the new loss function and the new negative sampler in their recommendation services. Nobody would be put at disadvantage from this research. The practical recommendation service usually adopt the ensemble of many recommendation models, so any single algorithm does not lead to any serious consequences of user experiences.

## Acknowledgements

The work was supported by grants from the National Natural Science Foundation of China (No. 61976198, 62022077, 61922073 and U1605251), Municipal Program on Science and Technology Research Project of Wuhu City (No. 2019yf05), and the Fundamental Research Funds for the Central Universities.

## Footnotes

[2]**Amazon**: http://jmcauley.ucsd.edu/data/amazon; **MovieLens**: https://grouplens.org/datasets/movielens; **CiteULike**: https://github.com/js05212/citeulike-t; **Gowalla**: http://snap.stanford.edu/data/loc-gowalla.html; **Echonest**: https://blog.echonest.com/post/3639160982/million-song-dataset

[3]https://github.com/guoguibing/librec

[4]https://github.com/lyst/lightfm

[5]https://github.com/changun/CollMetric

[6]https://github.com/geek-ai/irgan

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
