[Supplementary Material]

# Supplementary Material for the Paper "Sampling-Decomposable Generative Adversarial Recommender"

**Binbin Jin**[†], **Defu Lian**[†§*], **Zheng Liu**[‡], **Qi Liu**[†§] **Jianhui Ma**[†], **Xing Xie**[‡] , **Enhong Chen**[†§*]
[†]School of Computer Science and Technology, University of Science and Technology of China
[§]School of Data Science, University of Science and Technology of China
[‡]Microsoft Research Asia
bb0725@mail.ustc.edu.cn, {liandefu, qiliuql, jianhui, cheneh}@ustc.edu.cn,
{zhengliu, xingx}@microsoft.com

## Appendix

In the appendix, we start from the proofs of theorem 2.1 and theorem 2.2 in section A. Then, we prove the correctness of proposition 2.2 and proposition 2.3 in section B. After that, the detailed derivation of our proposed loss is provided in section C. At last, the sensitivity of some important parameters is discussed in section D.

## A  Proofs of Theorems

Before providing the proofs of the theorems, we restate some important notations first. In the following, denote by $\mathcal{C}$ the set of $N$ contexts, $\mathcal{I}$ the set of $M$ items and $\mathcal{I}_c$ interacted items in a context $c$. The objective function of IRGAN is as follows:

$$\min_G \max_D \mathcal{J}(D, G) = \sum_{c \in \mathcal{C}} E_{i \sim P_{\text{true}}(\cdot|c)} \log D(i|c) + E_{j \sim P_G(\cdot|c)} \log \left(1 - D(j|c)\right),$$

where $P_{\text{true}}(\cdot|c)$ is an underlying true relevance distribution over candidate items and $P_G(\cdot|c)$ is a probability distribution used to generate negative samples. $D(i|c) = \sigma(g_\phi(c, i)) = \frac{1}{1 + \exp(-g_\phi(c, i))}$ estimates the probability of preferring item $i$ in a context $c$.

**Theorem A.1** (**Theorem 2.1**). *Assume $G$ has enough capacity. Given the discriminator $D$, $\min_G \mathcal{J}(D, G)$ yields the optimum of $G$ as follows*

$$P_{G^\star}(\cdot|c) = \text{one-hot}(\arg\max_i(g_\phi(c, i))).$$

*Proof.* From the definition of $\mathcal{J}(D, G)$, when the discriminator $D$ is fixed, the first expectation is independent to $G$ so that it can be omitted when minimizing $\mathcal{J}$ w.r.t. $G$. Thus, the objective function is equivalent to

$$\min_G \sum_{c \in \mathcal{C}} E_{j \sim P_G(\cdot|c)} \log(1 - D(j|c)).$$

Let $h_c(j) = \log(1 - D(j|c))$, $j^\star = \arg\min_j h_c(j)$ so that $h_c(j^\star) \leq h_c(j), \forall j \in \mathcal{I}$. Note that $h_c(j)$ is a decreasing function w.r.t. $g_\phi(c, j)$, so $j^\star = \arg\min_j h_c(j) = \arg\max_j g_\phi(c, j)$. Then, in a context $c$, we have:

$$E_{j \sim P_G(\cdot|c)} h_c(j) = \sum_{j \in \mathcal{I}} P_G(j|c) h_c(j) \geq \sum_{j \in \mathcal{I}} P_G(j|c) h_c(j^\star) = h_c(j^\star) = \sum_{j \in \mathcal{I}} \text{one-hot}(j^\star) h_c(j).$$

---

[*]Defu Lian and Enhong Chen are corresponding authors.

Therefore, the optimum of $G$ follows $P_{G^*}(\cdot|c) = \text{one-hot}(\arg\max_i(g_\phi(c,i)))$. □

**Theorem A.2 (Theorem 2.2).** *Assume $G$ has enough capacity and let $f_c(i) = -\log(1 - D(i|c)) = \log(1 + \exp(g_\phi(c,i)))$. Given the discriminator $D$, $\min_G \mathcal{J}(D,G) - T \cdot \mathcal{H}(P_G(\cdot|c))$ yields the optimum of $G$ as follows*

$$P_{G_T^\star}(i|c) = \frac{\exp(f_c(i)/T)}{\sum_j \exp(f_c(j)/T)}.$$

*Proof.* From the definition of $\mathcal{J}(D,G)$, when the discriminator $D$ is fixed, the first expectation is independent to $G$ so that it can be omitted when minimizing $\mathcal{J}$ w.r.t. $G$. Thus, the objective function is equivalent to

$$\min_G \sum_{c \in \mathcal{C}} \left( E_{j \sim P_G(\cdot|c)} \log(1 - D(j|c)) - T \cdot \mathcal{H}(P_G(\cdot|c)) \right),$$

where $\mathcal{H}(P_G(\cdot|c)) = -\sum_{i \in \mathcal{I}} P_G(i|c) \log P_G(i|c)$ is the entropy regularization controlled by the temperature $T$. For simplicity of writing, regarding a certain context $c$, let $\boldsymbol{x} = [x_0, x_1, ..., x_{M-1}]$ where $x_j = P_G(j|c)$ and $\boldsymbol{y} = [y_0, y_1, ..., y_{M-1}]$ where $y_j = \log(1 - D(j|c)) = -f_c(j)$. Then, we can formalize the primal problem for the context $c$ as follows:

$$\min_{\boldsymbol{x}} \quad \boldsymbol{x}^\top \boldsymbol{y} + T \sum_{k=0}^{M-1} x_k \log x_k,$$

$$s.t. \quad \sum_{k=0}^{M-1} x_k = 1,$$

$$- x_k < 0, \quad k = 0, ..., M-1.$$

This is a constrained optimization problem, so we can define the Lagrange $\mathcal{L}$ as follows:

$$\mathcal{L}(\boldsymbol{x}, \alpha_0, ..., \alpha_{M-1}, \beta) = \boldsymbol{x}^\top \boldsymbol{y} + T \sum_{k=0}^{M-1} x_k \log x_k - \sum_{k=0}^{M-1} \alpha_k x_k + \beta\left(\sum_{k=0}^{M-1} x_k - 1\right),$$

where $\{\alpha_0, \alpha_1, ..., \alpha_{M-1}, \beta\}$ are the Lagrange multipliers. Obviously, the primal problem is convex and the equality constraint is affine so that the **KKT** conditions are also sufficient for the points to be primal and dual optimal. Suppose $\hat{x}_k, \hat{\alpha}_k, \hat{\beta}$ are any primal and dual optimal, then, we have:

$$\begin{cases} \frac{\partial \mathcal{L}}{\partial \hat{x}_k} = y_k + T(\log \hat{x}_k + 1) - \hat{\alpha}_k + \hat{\beta} = 0, & k = 0, ..., M-1 \\ \hat{\alpha}_k \hat{x}_k = 0, & k = 0, ..., M-1 \\ \hat{\alpha}_k \geq 0, & k = 0, ..., M-1 \\ \hat{x}_k > 0, & k = 0, ..., M-1 \\ \sum_{k=0}^{M-1} \hat{x}_k - 1 = 0 \end{cases}$$

$$\Rightarrow \begin{cases} \hat{\alpha}_k = 0, & k = 0, ..., M-1 \\ \hat{x}_k = \exp(\frac{-y_k - \hat{\beta} - T}{T}), & k = 0, ..., M-1 \\ \sum_{k=0}^{M-1} \exp(\frac{-y_k - \hat{\beta} - T}{T}) = 1 \end{cases}$$

$$\Rightarrow \hat{x}_k = \frac{\exp(-y_k/T)}{\sum_{k'=0}^{M-1} \exp(-y_{k'}/T)}, \quad k = 0, ..., M-1$$

Since $x_k = P_G(k|c)$ and $y_k = -f_c(k)$, the optimal solution $P_{G_T^\star}(i|c) = \frac{\exp(f_c(i)/T)}{\sum_j \exp(f_c(j)/T)}$. □

# B  Proofs of Propositions

Here, we also restate some important notations first. $\mathcal{S}_c$ is a set of samples drawn from the generator $Q_G(\cdot|c)$ in the context $c$. Our approximated loss and its variance are as follows:

$$\mathcal{V}_T(D,\mathcal{S}) = \sum_{c\in\mathcal{C}}\left(-\frac{1}{|\mathcal{I}_c|}\sum_{i\in\mathcal{I}_c}\log D(i|c) - \sum_{j\in\mathcal{S}_c} w_{cj}\log\left(1-D(j|c)\right)\right),$$

$$w_{cj} = \frac{\exp\left(f_c(j)/T - \log\tilde{Q}_G(j|c)\right)}{\sum_{i\in\mathcal{S}_c}\exp\left(f_c(i)/T - \log\tilde{Q}_G(i|c)\right)},$$

$$\text{Var}\left(\mathcal{V}_T(D,\mathcal{S})\right) = \sum_{c\in\mathcal{C}}\frac{1}{|\mathcal{S}_c|}\sum_{i\in\mathcal{I}}\frac{P_{G_T^\star}(i|c)^2(f_c(i)-\mu_c)^2}{Q_G(i|c)},$$

where $\mathcal{S} = \bigcup_{c\in\mathcal{C}}\mathcal{S}_c$ and $\tilde{Q}_G(j|c)$ is the unnormalized $Q_G(j|c)$.

**Proposition B.1 (Proposition 2.2).** *Var* $\left(\mathcal{V}_T(D,\mathcal{S})\right) \geq \sum_{c\in\mathcal{C}}\frac{1}{|\mathcal{S}_c|}\mathbb{E}_{i\sim P_{G_T^\star}(\cdot|c)}(|f_c(i)-\mu_c|)^2$, *where the equality holds if* $Q_G(i|c)\propto P_{G_T^\star}(i|c)|f_c(i)-\mu_c|$.

*Proof.* According to Cauchy–Schwarz inequality, let $X$ and $Y$ be random variables, then we have the following inequality

$$E(X^2)E(Y^2) \geq |E(XY)|^2.$$

Now, let $Y \equiv 1$ and suppose all contexts are IID, so we have:

$$\begin{aligned}
\text{Var}\left(\mathcal{V}_T(D,\mathcal{S})\right) &= \sum_{c\in\mathcal{C}}\frac{1}{|\mathcal{S}_c|}\sum_{i\in\mathcal{I}}\frac{P_{G_T^\star}(i|c)^2(f_c(i)-\mu_c)^2}{Q_G(i|c)}\\
&= \sum_{c\in\mathcal{C}}\frac{1}{|\mathcal{S}_c|}E_{i\sim Q_G(i|c)}\left(\left|\frac{P_{G_T^\star}(i|c)(f_c(i)-\mu_c)}{Q_G(i|c)}\right|^2\right)\\
&\geq \sum_{c\in\mathcal{C}}\frac{1}{|\mathcal{S}_c|}E_{i\sim Q_G(i|c)}\left(\left|\frac{P_{G_T^\star}(i|c)(f_c(i)-\mu_c)}{Q_G(i|c)}\right|\right)^2\\
&= \sum_{c\in\mathcal{C}}\frac{1}{|\mathcal{S}_c|}E_{i\sim P_{G_T^\star}(i|c)}(|f_c(i)-\mu_c|)^2
\end{aligned}$$

When $Q_G(i|c)\propto P_{G_T^\star}(i|c)|f_c(i)-\mu_c|$, let

$$Q_G(i|c) = \frac{P_{G_T^\star}(i|c)|f_c(i)-\mu_c|}{\sum_{j\in\mathcal{I}}P_{G_T^\star}(j|c)|f_c(j)-\mu_c|} = \frac{P_{G_T^\star}(i|c)|f_c(i)-\mu_c|}{E_{j\sim P_{G_T^\star}(\cdot|c)}|f_c(j)-\mu_c|}.$$

Then, the variance becomes

$$\begin{aligned}
\text{Var}\left(\mathcal{V}_T(D,\mathcal{S})\right) &= \sum_{c\in\mathcal{C}}\frac{1}{|\mathcal{S}_c|}\sum_{i\in\mathcal{I}}\frac{P_{G_T^\star}(i|c)^2(f_c(i)-\mu_c)^2}{Q_G(i|c)}\\
&= \sum_{c\in\mathcal{C}}\frac{1}{|\mathcal{S}_c|}\sum_{i\in\mathcal{I}}P_{G_T^\star}(i|c)|f_c(i)-\mu_c|E_{j\sim P_{G_T^\star}}|f_c(j)-\mu_c|\\
&= \sum_{c\in\mathcal{C}}\frac{1}{|\mathcal{S}_c|}E_{j\sim P_{G_T^\star}}|f_c(j)-\mu_c|E_{i\sim P_{G_T^\star}}|f_c(i)-\mu_c|\\
&= \sum_{c\in\mathcal{C}}\frac{1}{|\mathcal{S}_c|}E_{i\sim P_{G_T^\star}}(|f_c(i)-\mu_c|)^2.
\end{aligned}$$

$\square$

**Proposition B.2** (**Proposition 2.3**). *If $\forall c$, $\mathcal{S}_c$ drawn i.i.d from uniform$(\mathcal{I})$, then $w_{cj} = \frac{\exp(f_c(j)/T)}{\sum_{i \in \mathcal{S}_c} \exp(f_c(i)/T)}$ and $\forall 0 < T_1 < T_2 < +\infty$,*

$$\lim_{T \to +\infty} \mathcal{V}_T(D, \mathcal{S}) < \mathcal{V}_{T_2}(D, \mathcal{S}) < \mathcal{V}_{T_1}(D, \mathcal{S}) < \lim_{T \to 0} \mathcal{V}_T(D, \mathcal{S}).$$

*Proof.* To prove this proposition, we just have to prove that $\mathcal{V}_T(D, \mathcal{S})$ is a decreasing function w.r.t. $T$ $\forall T > 0$.

$$\frac{\partial \mathcal{V}_T(D, \mathcal{S})}{\partial T} = \sum_{c \in \mathcal{C}} \sum_{j \in \mathcal{S}_c} \frac{\partial w_{cj}}{\partial T} f_c(j).$$

Considering the second summation, for simplicity of writing, let $f_j = f_c(j)$ for a certain context $c$. Then, for a context $c$, we have

$$\sum_{j \in \mathcal{S}_c} \frac{\partial w_{cj}}{\partial T} f_j = \sum_{j \in \mathcal{S}_c} \frac{\exp(\frac{f_j}{T})(-\frac{f_j}{T^2})\left(\sum_{i \in \mathcal{S}_c} \exp(\frac{f_i}{T})\right) - \exp(\frac{f_j}{T})\left(\sum_{i \in \mathcal{S}_c} \exp(\frac{f_i}{T})(-\frac{f_i}{T^2})\right)}{\left(\sum_{i \in \mathcal{S}_c} \exp(\frac{f_i}{T})\right)^2} f_j$$

$$= \frac{1}{\left(\sum_{i \in \mathcal{S}_c} \exp(\frac{f_i}{T})\right)^2} \sum_{j \in \mathcal{S}_c} \sum_{i \in \mathcal{S}_c} \exp(\frac{f_i + f_j}{T})(\frac{f_i f_j - f_j^2}{T^2}).$$

When $i = j$, it is obvious that the addend equals $0$. Regarding the rest of addends, we can rearrange them into a set of pairs. Specifically, let $h(i, j) = \exp(\frac{f_i + f_j}{T})(\frac{f_i f_j - f_j^2}{T^2})$. $\forall i, j \in \mathcal{S}_c \wedge i \neq j$, $h(i, j) + h(j, i) = \exp(\frac{f_i + f_j}{T})(\frac{-(f_i - f_j)^2}{T^2}) < 0$. Therefore, we have $\sum_{j \in \mathcal{S}_c} \frac{\partial w_{cj}}{\partial T} f_j < 0$ so that $\frac{\partial \mathcal{V}_T(D, \mathcal{S})}{\partial T} < 0$. In other words, $\mathcal{V}_T(D, S)$ is a decreasing function w.r.t. $T$. $\square$

## C  Derivation of the Proposed Objective Function

Here, we illustrate the detailed derivation of our approximated loss for learning the discriminator. For each context $c$, considering items in $\mathcal{I}_c$ are observed data sampled from $P_{true}(\cdot|c)$ which are IID, items in $\mathcal{S}_c$ are sampled from $Q_G(\cdot|c)$. Then, we have:

$$E_{i \sim P_{true}(\cdot|c)} \log D(i|c) \approx \frac{1}{|\mathcal{I}_c|} \sum_{i \in \mathcal{I}_c} \log D(i|c),$$

$$E_{j \sim P_{G_T^\star}(\cdot|c)} \log(1 - D(j|c)) \approx \frac{1}{|\mathcal{S}_c|} \sum_{j \in \mathcal{S}_c} \frac{P_{G_T^\star}(j|c)}{Q_G(j|c)} \log(1 - D(j|c)),$$

$$P_{G_T^\star}(j|c) = \frac{\exp(f_c(j)/T)}{\sum_{i \in \mathcal{I}} \exp(f_c(i)/T)}.$$

In particular, the normalization constant of $P_{G_T^\star}(\cdot|c)$ (denoted as $Z_{G_T^\star}$) can be approximated by the samples $\mathcal{S}_c$ as:

$$Z_{G_T^\star} = \sum_{i \in \mathcal{I}} \exp(f_c(i)/T) = E_{i \sim Q_G(\cdot|c)} \frac{\exp(f_c(i)/T)}{Q_G(\cdot|c)} \approx Z_Q \frac{1}{|\mathcal{S}_c|} \sum_{i \in \mathcal{S}_c} \exp(f_c(i)/T - \log \tilde{Q}_G(i|c)),$$

where $\tilde{Q}_G(i|c)$ is the unnormalized $Q_G(i|c)$ such that $\tilde{Q}_G(i|c) = Z_Q Q_G(i|c)$. Then, we can approximate $\frac{P_{G_T^\star}(j|c)}{Q_G(j|c)}$ as follows:

$$\frac{P_{G_T^\star}(j|c)}{Q_G(j|c)} \approx \frac{\exp(f_c(j)/T)}{Q_G(j|c) Z_Q \frac{1}{|\mathcal{S}_c|} \sum_{i \in \mathcal{S}_c} \exp(f_c(i)/T - \log \tilde{Q}_G(i|c))}$$

$$= \frac{\exp(f_c(j)/T - \log \tilde{Q}_G(j|c))}{\frac{1}{|\mathcal{S}_c|} \sum_{i \in \mathcal{S}_c} \exp(f_c(i)/T - \log \tilde{Q}_G(i|c))}.$$

To sum up:

$$\mathcal{J}(D, G_T^\star) = \sum_{c \in \mathcal{C}} -\mathbb{E}_{i \sim P_{\text{true}}(\cdot|c)} \log D(i|c) - \mathbb{E}_{j \sim P_{G_T^\star}(\cdot|c)} \log\left(1 - D(j|c)\right)$$

$$= \sum_{c \in \mathcal{C}} -\mathbb{E}_{i \sim P_{\text{true}}(\cdot|c)} \log D(i|c) - \mathbb{E}_{j \sim Q_G(\cdot|c)} \frac{P_{G_T^\star}(j|c)}{Q_G(j|c)} \log\left(1 - D(j|c)\right)$$

$$\approx \mathcal{V}_T(D, \mathcal{S}) = \sum_{c \in \mathcal{C}} \left( -\frac{1}{|\mathcal{I}_c|} \sum_{i \in \mathcal{I}_c} \log D(i|c) - \sum_{j \in \mathcal{S}_c} w_{cj} \log\left(1 - D(j|c)\right) \right),$$

$$w_{cj} = \frac{\exp\left(f_c(j)/T - \log \tilde{Q}_G(j|c)\right)}{\sum_{i \in \mathcal{S}_c} \exp\left(f_c(i)/T - \log \tilde{Q}_G(i|c)\right)}.$$

(a) Embedding size $K$    (b) $|\mathcal{S}_c|$ for learning discriminator    (c) $|\mathcal{S}_c|, |\mathcal{S}_i|$ for learning generator

Figure 1: Effects of different hyper parameters.

## D   Parameter Sensitivity

Here, we explore the sensitivity of some important parameters including the embedding size, the number of negative samples for the discriminator, and the number of samples for the generator. We report the results on two datasets (i.e., CiteULike and Gowalla). For the other datasets, similar observations can be found.

Figure 1(a) demonstrates the effects of the embeddings size (i.e., $K$). We vary the dimension of user and item embeddings in the set $\{16, 32, 64, 128, 256\}$. We can observe when the embedding size increases, the performance improves quickly at first and then slows down. Considering that when the embedding size becomes larger, the training and inference stages will spend more time. Thus, it is significant to choose an appropriate size in practice.

Figure 1(b) shows the effects of the number of item sample set for learning the discriminator. It has a similar tendency to the embedding size. We vary the number of negative samples from 1 to 20 with a step 5. The results demonstrate that when $\mathcal{S}_c$ is larger than 5, the improvements is limited, and even a slight drop. This observation implies feeding more negative samples with weight scores can improve the recommendation performance.

Figure 1(c) reports the effects of the number of item and context sample set for learning the generator. We set $|\mathcal{S}_c| = |\mathcal{S}_i|$ and vary the numbers in the set $\{8, 16, 32, 64, 128\}$. We can find SD-GAR is not sensitive to this hyper-parameter. This observation ensures that it is effective to utilize sampling techniques for approximation when updating the generator. In addition, this conclusion also reveals the computation cost can be further reduced by cutting down the sample number.