[Reviews · NeurIPS 2020]

Review 1

Summary and Contributions: This paper overcomes the training difficulty of existing GAN-based recommender models, IRGAN, and proposes a new recommender model, called sampling-decomposable generative adversarial recommender (SD-GAR), which overcomes the divergence of the generator by self-normalized importance sampling and decompose the sampling method from training the generator. Furthermore, it utilizes the ALS-based method for efficient training. Experimental results show that SD-GAR outperforms IRGAN and other baseline models.

Strengths: 1. This paper effectively analyzes the limitation of the existing GAN-based recommender model, IRGAN. 2. This paper is well-written and shows a thorough analysis and proof of the proposed model. 3. It shows an extensive experimental evaluation with various datasets.

Weaknesses: 1. It is necessary to show more thorough ablation studies such as the effect of the loss function, the effect of different sampling methods, and the effect of training methods. 2. It would be better to compare the state-of-the-art GAN-based recommender model. Please refer to the following reference. - Chae at el., "CFGAN: A Generic Collaborative Filtering Framework based on Generative Adversarial Networks," CIKM 2018 https://dl.acm.org/doi/10.1145/3269206.3271743 3. It would better to show a smaller size N of NDCG@N, e.g., 10 or 25.

Correctness: The proposed method is correct. However, it would be better to show a more thorough evaluation, including a state-of-the-art model, i.e., CFGAN.

Clarity: This paper is well-written, and the organization of this paper is clear.

Relation to Prior Work: It would welcome discussing the recent GAN-based recommender model. Also, it needs to include more recent state-of-the-art recommender models, such as graph convolution-based recommender models and autoencoder-based recommender models. Please refer to the following papers. - Xiangnan He et al., "LightGCN: Simplifying and Powering Graph Convolution Network for Recommendation," SIGIR 2020 https://arxiv.org/abs/2002.02126 - Dawen Liang et al., "Variational Autoencoders for Collaborative Filtering," WWW 2018 https://arxiv.org/pdf/1802.05814.pdf

Reproducibility: Yes

Additional Feedback: After rebuttal, I found that the experimental results in CFGAN are too lower than other papers, including CFGAN and IRGAN. Please refer to the following papers. Chae at el., "CFGAN: A Generic Collaborative Filtering Framework based on Generative Adversarial Networks," CIKM 2018 https://dl.acm.org/doi/pdf/10.1145/3269206.3271743 Zhou et al,, "Recommendation via collaborative autoregressive flows," Neural Networks 2020 https://www.sciencedirect.com/science/article/pii/S0893608020300873 I would recommend that the authors check the evaluation setup and hyperparameter tuning.


Review 2

Summary and Contributions: The paper revisits an important work -- IRGAN, which uses GAN to sample training examples for IR problems, and identifies two problems (with theoretical analysis) of it: deviation due to the limited capacity, and high sampling complexity. Moreover, they proposed solutions (new objective and decomposable sampling) to nicely address these problems. The experiments are quite extensive with various strong baselines, the performance improvement on both ranking performance and time consumption is significant, and the setting looks convincing to me. Lastly, the paper is well written in all the aspects (intuition, theory, experiments), and easy to follow. I consider this is a solid improvement on IRGAN which makes it much better and more efficient.

Strengths: - Nicely identify, analyze, address the two problems in IRGAN with reasonable methods. - Extensive experiments and impressive performance - Well written

Weaknesses: - IRGAN is designed for various IR tasks while here only focus on recommendation. However, the results on recommendation are solid to me, so I guess it may work well on other tasks, though it'd be better to have the results as well.

Correctness: Yes

Clarity: Good

Relation to Prior Work: Good

Reproducibility: Yes

Additional Feedback: The rebuttal mostly addresses my concerns, the supplementary results make the paper more convincing to me. Hence I maintain my rating of accept.


Review 3

Summary and Contributions: This paper analyzed well-known GAN based information retrieval framework IRGAN in the recommendation setting. It proposed multiple interesting modifications that significantly improve its training efficiency and scalability for recommendation tasks. Specifically, the paper first pointed out two problems of IRGAN: (1) simple GAN objective could cause the optimal negative sampler biases to extreme cases (delta distributions), (2) Sampling from the optimal negative sampler is computationally expensive. For addressing (1), the paper proposed to add an entropy regularization that smooth the negative sampler distribution (optimal). For addressing (2), the paper suggested using self-normalized important sampling to approximate optimal negative sampler found in (1), where sampling from proposed distribution could be decomposed into two-step categorical sampling. Further, the paper described a strategy for learning proposed distribution by minimizing estimation variance through a constrained optimization. I have read author feedback and I believe my review conclusion no need to change.

Strengths: The idea of the paper is novel and exciting. The paper writing quality is good, but some parts need further clarification.

Weaknesses: The experiment part is suspicious in terms of baseline as well as metric choice. For baselines, there are many strong algorithms proposed and used in practice, such as classic WRMF (still the winner of recommendation challenges RecSys 2018) published in 2008, VAE-CFs, Neural Collaborative Filterings~(NCFs). Why only BPR, CML, etc., that has been repeatedly outperformed on the benchmark datasets? The authors probably need to justify the baseline choice a bit. For metric, NDCG@50 is over too much for the datasets used in this paper. There are rare users who have around 50 historical interaction records. It is hard to justify the performance of the proposed algorithm. Over 10% of performance improvement on benchmark datasets is especially hard to believe after years of many good algorithms published.

Correctness: yes

Clarity: Yes, the paper is in good shape. However, as stated, some sections need more clarification

Relation to Prior Work: yes

Reproducibility: Yes

Additional Feedback: Suggestions: 1. The paper is hard to follow due to many mathematical descriptions but omitting connections among contents. For example, after describing important sampling in proposition 2.1, the paper directly turns into explaining the variance of proposed distribution without mentioning why we want to look at it. It is better to use one or two-sentence to state that reducing the variance of estimator could help optimize the proposed distribution beforehand. 2. The connection between equation 7 and proposition 2.2 is also a bit off. It is better to state why maximizing equation 7 is equivalent to minimizing estimator variance directly. Proposition 2.2 does not show/indicate the optimization objective explicitly. 2. Equation 2, A distribution equals to a one-hot vector is odd, maybe Cat(K, one-hot(...)) 3. Conduct a comprehensive comparison to the SOTA models published in the last three years, if possible.


Review 4

Summary and Contributions: This paper points out the limitations of previous generative-retrieval recommender systems; 1) sampling items from the generator is time-consuming and 2) the discriminator performs poor in top-k item recommendation, which would be caused by the divergence between the generator and the optimum. To tackle these challenges, this paper proposes Sampling-Decomposable Generative Adversarial Recommender (SD-GAR) which compensates the divergence between the generator and the optimum as well as decomposes the generator to reduce the sampling cost of the generator.

Strengths: 1.This paper provides the theoretical analysis on the generative-retrieval recommender system. The authors theoretically show its optimal sampling distribution and the approximation to the optimum. 2. This paper proposes the closed-form solution for optimization of the generator as well as the new sampling-decomposable generator. 3. The proposed algorithm achieves remarkable improvements over the state-of-the art competitor in terms of both scalability and recommendation accuracy.

Weaknesses: 1. Some related papers are not investigated and not discussed in this paper. The authors are strongly recommended to include the survey on the related work. 2. Experiments are not solid: 1) unclear hyperparameter setups, 2) weak analysis on the divergence of G (or poor performance of D).

Correctness: In overall, the claims of this paper are technically sound. However, the empirical evaluation for the proposed method is not sufficient to validate the claims of the paper. The optimal hyperparameter values are not thoroughly searched (or not mentioned at all) for the competitors. For example, the L2-regularization coefficient in BPR and the margin size in CML largely affects their final performances, thus they should be specified. Only a single metric is used for evaluation (i.e., NDCG@50). More ranking performances should be reported, including MRR and MAP. In case that the size of the latent dimension is small, comparing Gan-like methods with traditional latent factor models is unfair because the number of parameters in Gan-like methods is much larger than that in latent factor models.

Clarity: The paper is well written in overall, but some parts should be further clear: First, the authors claimed that the discriminator D (rather than the generator G) should be considered as a recommender due to the data sparsity issue in the generator G. However, the authors also repeatedly mentioned that the discriminator D shows poor performances in top-k recommendation, which makes the readers confused. Second, some typos (e.g., index u in eq. 4) should be corrected.

Relation to Prior Work: Some related papers are not investigated and not discussed in this paper. [1] CFGAN: A Generic Collaborative Filtering Framework based on Generative Adversarial Networks. [2] Adversarial Binary Collaborative Filtering for Implicit Feedback.

Reproducibility: No

Additional Feedback:

[Author Response · NeurIPS 2020]

Table 1: Comparison results with baselines on three datasets w.r.t. four metrics.

| | CiteULike | | | | MovieLens | | | | Gowalla | | | |
|---|---|---|---|---|---|---|---|---|---|---|---|---|
| | P@10 | NDCG@10 | NDCG@50 | MRR | P@10 | NDCG@10 | NDCG@50 | MRR | P@10 | NDCG@10 | NDCG@50 | MRR |
| SD-GAR | **0.0366** | **0.0997** | **0.1365** | **0.1409** | **0.1311** | **0.2254** | **0.3134** | **0.3711** | **0.0652** | **0.1120** | **0.1620** | **0.2366** |
| CFGAN | 0.0031 | 0.0091 | 0.0109 | 0.0147 | 0.0471 | 0.0684 | 0.1076 | 0.1425 | 0.0009 | 0.0026 | 0.0031 | 0.0066 |
| CFGAN$^\star$ | 0.0070 | 0.0205 | 0.0296 | 0.0345 | 0.0661 | 0.1130 | 0.1660 | 0.2101 | 0.0207 | 0.0336 | 0.0428 | 0.0873 |
| VAECF | 0.0206 | 0.0519 | 0.0820 | 0.0761 | 0.0764 | 0.1300 | 0.2251 | 0.2214 | 0.0424 | 0.0735 | 0.1172 | 0.1583 |
| U-GAR | 0.0347 | 0.0897 | 0.1269 | 0.1305 | 0.1148 | 0.1920 | 0.2764 | 0.3251 | 0.0605 | 0.1048 | 0.1490 | 0.2265 |
| BCE | 0.0298 | 0.0753 | 0.1071 | 0.1121 | 0.0984 | 0.1650 | 0.2362 | 0.2913 | 0.0504 | 0.0870 | 0.1261 | 0.1927 |

**Q1**: It is better to compare the SOTA methods (e.g. CFGAN, VAECF, WRMF, NCF), show results on various metrics.

**A1**: We perform a comparison with CFGAN and VAECF with source codes released by the original authors. For
CFGAN, both the author's suggested setting and our optimally tuned setting (CFGAN$^\star$) are tested. For VAECF, the
parameters are optimally tuned following the original author's suggestion. (CML is extensively verified to be better than
WRMF; while NCF is not a fair comparison for MF-based approaches, as it is an ensemble of MF and NN.) We also
report performances on a wide spectrum of extra metrics, like MRR, P@10 and NDCG@10. It can be observed that our
approach, SD-GAR, significantly outperforms these additional baselines, which further verifies our effectiveness.

**Q2**: It is necessary to show more thorough ablation studies such as the effect of the loss function, the effect of different
sampling methods, and the effect of training methods.

**A2**: SD-GAR's advantages can be fully explained with two ablation studies. 1) U-GAR replaces the proposed sampler
with uniform sampler, and 2) BCE substitutes the loss function with binary cross entropy. Given the substantial
performance gain in Table 1, we may conclude that 1) SD-GAR's sampling strategy is much better than conventional
uniform sampling; 2) the proposed loss function is more effective than commonly used BPR (already reported) and
BCE. Note that the alternative training method, i.e., policy gradient, is not compared, as it calls for normalization over
all items of $\boldsymbol{y}_{\cdot k}$ for each $k$, which is so time-consuming that training can not be completed within a short rebuttal period
with straightforward implementation. The improved performance of SD-GAR already verifies the effectiveness of the
proposed training method; we will further study how to implement policy gradient efficiently in future work.

**Q3**: It is better to state the connection from the proposition 2.1 to the variance of the estimator. In addition, it is better
to explain the relation between maximizing Eq. 7 and minimizing estimator variance.

**A3**: Following asymptotic unbiasedness of the estimator, we show its variance so as to provide guidance for variance
reduction, which can help optimize the proposal $Q$. We will follow reviewer's suggestion to connect them more
smoothly. According to Theorem 2.2, with entropy regularization, the optimum of Eq (7) is achieved when $\boldsymbol{x}_c^\top \boldsymbol{y}_i \propto$
$\exp\left(P_{G_T^\star}(i|c)|f_c(i) - \mu_c|/T\right)$, which is approximately $\propto P_{G_T^\star}(i|c)|f_c(i) - \mu_c|$ when $T$ is comparatively large.

**Q4**: The optimal hyperparameter values are not thoroughly declared for the competitors, e.g., the L2-regularization
coefficient in BPR and the margin size in CML.

**A4**: The parameters for the baselines are optimally tuned within the following scopes. For all the competitors, the
L2-regularization coefficient is tuned over {0.01, 0.03, 0.05}. The margin size in CML is tuned over {0.5, 1.0, 1.5, 2.0}.
Other hyperparameters, e.g., the embedding size and the number of negative samples are set to the same as SD-GAR.

**Q5**: In case that the size of the latent dimension is small, comparing Gan-like methods with traditional latent factor
models is unfair because the number of parameters in Gan-like methods is much larger than that in latent factor models

**A5**: In fact, our recommender only uses D, which is of the same size as other latent factor models, for recommendation.
While G is only a sampler which is used to help with the training of D, it does not take part in the prediction of
recommendation score. Therefore, the comparison is fair for all the reported methods.

**Q6**: The authors claimed that the discriminator D (rather than the generator G) should be considered as a recommender
due to the data sparsity issue in the generator G. However, the authors also repeatedly mentioned that the discriminator
D shows poor performances in top-k recommendation, which makes the readers confused.

**A6**: Generally speaking, *the discriminator (D) rather than generator (G) is more suitable for making recommendation,*
*because D learns directly from training data, whereas G merely learns from samples drawn from the generator*
*distribution; besides, the learning of G is guided by D, which can be not reliable.* Unfortunately, D is not well trained
in IRGAN, as G is pretrained, which becomes more likely to generate "hard cases", and in return harms the training
performance of D in the initial stage. Our proposed framework does not have such a limitation: the generator is not
required to be initiated highly accurate. Instead, the accuracy of G is improved simultaneously with D: when D is
initialized, G only shows it with easy cases; when D improves, G will be enhanced as well and gradually present more
difficult cases. As a result, G will always contribute to D's training performance without introducing any side effect.

Finally, we will follow reviewers' other suggestions to make discussion on related works, e.g., LightGCN, CFGAN,
VAECF, etc, and revise all the typos and unclear expressions.

[Meta-Review · NeurIPS 2020]

This paper received overall positive scores. All the reviewers agree that the theoretical analysis on GAN-based recommender models is the main strength of the paper. One reviewer raised some concern regarding the metrics obtained by the baseline being too low. I suggest that the authors carefully check the evaluation procedure, as well as incorporate the feedback from the reviews in the revision.